# Dividing attention during the Timed Up and Go enhances associations of several subtask performances with MCI and cognition

Victoria N. Poole[1,2]*, Robert J. Dawe[1,3], Melissa Lamar[1,4], Michael Esterman[5,6], Lisa Barnes[1,4,7], Sue E. Leurgans[1,7,8], David A. Bennett[1,7], Jeffrey M. Hausdorff[1,2,9,10], Aron S. Buchman[1,7]

1 Rush Alzheimer's Disease Research Center, Rush University Medical Center, Chicago, Illinois, United States of America, 2 Department of Orthopedic Surgery, Rush University Medical Center, Chicago, Illinois, United States of America, 3 Department of Diagnostic Radiology and Nuclear Medicine, Rush University Medical Center, Chicago, Illinois, United States of America, 4 Department of Psychiatry and Behavioral Sciences, Rush University Medical Center, Chicago, Illinois, United States of America, 5 National Center for PTSD & Boston Attention and Learning Laboratory, VA Boston Healthcare System, Boston, Massachusetts, United States of America, 6 Department of Psychiatry, Boston University School of Medicine, Boston, Massachusetts, United States of America, 7 Department of Neurological Sciences, Rush University Medical Center, Chicago, Illinois, United States of America, 8 Department of Preventive Medicine, Rush University Medical Center, Chicago, Illinois, United States of America, 9 Center for the Study of Movement, Cognition, and Mobility, Tel Aviv Sourasky Medical Center, Tel Aviv, Israel, 10 Sagol School of Neuroscience and Department of Physical Therapy, Sackler Faculty of Medicine, Tel Aviv University, Tel Aviv, Israel

* victoria_poole@rush.edu

**Data Availability Statement:** More information regarding obtaining these data for research use can be found at the RADC Research Resource Sharing Hub (www.radc.rush.edu).

## Abstract

We tested the hypothesis that dividing attention would strengthen the ability to detect mild cognitive impairment (MCI) and specific cognitive abilities from Timed Up and Go (TUG) performance in the community setting. While wearing a belt-worn sensor, 757 dementia-free older adults completed TUG during two conditions, with and without a concurrent verbal serial subtraction task. We segmented TUG into its four subtasks (i.e., walking, turning, and two postural transitions), and extracted 18 measures that were summarized into nine validated sensor metrics. Participants also underwent a detailed cognitive assessment during the same visit. We then employed a series of regression models to determine the combinations of subtask sensor metrics most strongly associated with MCI and specific cognitive abilities for each condition. We also compared subtask performances with and without dividing attention to determine whether the costs of divided attention were associated with cognition. While slower TUG walking and turning were associated with higher odds of MCI under normal conditions, these and other subtask associations became more strongly linked to MCI when TUG was performed under divided attention. Walking and turns were also most strongly associated with executive function and attention, particularly under divided attention. These differential associations with cognition were mirrored by performance costs. However, since several TUG subtasks were more strongly associated with MCI and cognitive abilities when performed under divided attention, future work is needed to determine how instrumented dual-task TUG testing can more accurately estimate risk for late-life cognitive impairment in older adults.

**Funding:** This study was supported by the National Institutes of Health (NIH; [grants: K01AG064044 to VNP, K25AG061254 to RJD, R01AG022018 to LLB, R01AG017917 to DAB, R01NS078009 to ASB, R01AG056352 to ASB]; the Illinois Department of Public Health to DAB; and the Robert C. Borwell Endowment Fund to DAB. The funders had no role in study design, data collection and analysis, decision to publish, or preparation of the manuscript.

**Competing interests:** The authors have declared that no competing interests exist.

## Introduction

The Timed Up and Go (TUG) test is widely used to assess mobility and risk of falls in older adults by geriatricians and aging researchers alike [1]. Although it was originally introduced as a test of mobility, several studies report that a longer overall TUG duration is associated with poorer cognition and faster cognitive decline in older adults [2, 3]. These findings are perhaps not surprising since voluntary actions require cognitive resources to plan, initiate, execute, and regulate the movements necessary for successful task completion [4, 5]. Nonetheless, it is not clear which components of the TUG drive its association with cognitive function and whether specific cognitive abilities are differentially related to individual TUG components.

In prior work, we quantified each of four distinct TUG subtasks (i.e., a transition from sit to stand, walking, turning, and transition from stand to sit) using a belt-worn sensor during testing in the community setting. We found that although TUG duration did not differ between those with and without mild cognitive impairment (MCI), altered postural control during the sit-to-stand transition, gait, and turns were associated with MCI status and lower cognition more generally [6]. Our other studies have shown that combinations of these TUG subtask metrics may improve the prediction of incident MCI and other adverse health outcomes [7]. However, given prior evidence that TUG components respond differently to divided attention challenges [8], dual-task performance of the TUG may uniquely affect individual subtasks, reveal how they rely on different cognitive abilities, and increase sensitivity to underlying cognitive impairment, including MCI.

In the current study, we sought to test the hypothesis that the associations between TUG subtask performances and MCI are made stronger when performed simultaneously with an unrelated task. We also hypothesized that individual subtask performances would be differentially linked to executive function and attentional abilities and affected by divided attention accordingly. To test these hypotheses, we utilized clinical, cognitive, and instrumented mobility data collected from more than 750 well-characterized dementia-free, ambulatory older adults participating in the Rush Memory and Aging Project (MAP) and Minority Aging Research Study (MARS).

## Methods

### Participants

Participants were drawn from two harmonized cohort studies of aging and dementia that share a common core of data collection procedures, resources, and staff at the Rush Alzheimer's Disease Center in Chicago, IL. The Rush Memory and Aging Project (MAP), which began in 1997, is a study of older adults primarily recruited from continuous care retirement communities and subsidized housing facilities in and around the Chicago metropolitan area [9]. The Minority Aging Research Study (MARS) began in 2004 and recruits older African Americans also living within the Chicagoland area [10]. Older adults without known dementia are eligible for these studies and agree to annual clinical testing in their individual homes. Participants in MAP also agree to brain donation at autopsy at the time of death; brain donation is optional for MARS participants. Both studies were approved by the Rush Institutional Review Board and written informed consent was obtained from all participants prior to testing.

### Study design

Although annual instrumented mobility testing began in 2011, dual-task Timed Up and Go (DT TUG) testing was added in 2016. The current study is a cross-sectional analysis of

participants who had completed both this protocol and a cognitive assessment during the same annual testing cycle and had no presence or history of dementia at that session. At the time of analyses, 818 participants had processed DT TUG data. Of this sample, 61 were excluded for having a history of dementia (see clinical diagnosis criteria below), leaving 757 participants for study analyses (MAP, n = 531; MARS, n = 226).

## Cognitive assessment

Annual cognitive testing was administered by a research assistant. Raw scores on 19 neuropsychological tests were converted to z-scores using the baseline evaluation of all participants across the parent studies. These scores were then averaged to obtain a global cognitive score, as used in prior publications [11]. To better isolate the contributions of attention and executive functioning to the global cognition summary score, we also constructed two measures from 6 of the 19 cognitive tests, as informed by prior literature (S1 Table) [12, 13]. These two metrics were compared to an episodic memory composite, which comprised 7 of the remaining tests and served as a reference domain anticipated not to have strong associations with mobility metrics.

## Clinical diagnosis of MCI and dementia

Clinical diagnoses were made in a three-step process, as previously described [14]. After the cognitive tests were scored, data were then reviewed by a neuropsychologist to rate cognitive impairment in each of five domains: episodic memory, semantic memory, working memory, perceptual speed, and visuospatial ability. The presence of dementia was then determined by an experienced clinician using the guidelines of the National Institute of Neurological and Communicative Disorders and Stroke and Alzheimer's Disease and Related Disorders Association [15]. Individuals who demonstrated evidence of cognitive impairment but not determined to have dementia were classified as having MCI, as previously described [16]. Individuals without a diagnosis of MCI or dementia were classified as having no cognitive impairment. These diagnostic determinations were used for research purposes only.

## Mobility testing and recording

In-home mobility testing included two TUG trials over 8 feet (i.e., 2.4 m) for each of two conditions: with and without a concurrent requirement to perform a serial 3's subtraction task. For the usual (normal) TUG, participants were instructed to stand up from a chair (i.e., sit-to-stand transition), walk 8 feet at their preferred pace (i.e., first 8-ft walk), return to the chair (i.e., mid-trial turn, second 8-ft walk), and sit (i.e., stand-to-sit transition). For DT TUG, participants were instructed to repeat the TUG while simultaneously subtracting threes from 100 aloud.

The testing session was recorded by a belt-worn Dynaport MoveTest (McRoberts B.V., the Netherlands) that was positioned over the lower back. This device contained a triaxial micro-electro-mechanical systems (MEMS) accelerometer and gyroscopic sensor to record acceleration and rotation, respectively, of the lower trunk at 100 Hz along each of three orthogonal directions. This state-of-the-art technology allowed us to quantify both spatiotemporal gait and trunk kinematics during the TUG procedure in the community setting. These time-stamped data were stored on onboard flash memory, then transferred to a secure RADC server for subsequent analyses.

### TUG subtask sensor metrics

Each continuous TUG trial recording was used to calculate the overall duration as well as 18 quantitative metrics, according to custom-developed algorithms [17, 18]. These metrics were transformed when appropriate to reduce skewness, averaged across trials for each condition, and z-scored relative to pooled parent study performance on the normal TUG. We then used principal component analyses (PCA) to reduce these data into interpretable subtask performances, similar to our prior work [19]. Since the underlying factor structures were similar across normal and dual-task TUG, we created the same nine mobility scores for each condition, according to S2 Table.

### Statistical analyses

To examine the associations of the individual TUG metrics with the odds of MCI, we first ran a series of logistic regression analyses considering overall TUG duration and each of the nine mobility scores (representing the four subtasks) as predictors of MCI for the normal and dual-task conditions separately. We then entered these measures into condition-specific forward selection stepwise procedures ($p < .05$ to enter, $p > .1$ to leave) to determine which metrics were most strongly associated with MCI at the time of mobility testing. To determine whether TUG performances were related to individual differences in cognitive abilities, we then examined linear associations with global cognition and three cognitive domains: attention, executive function, and episodic memory, and employed forward-selection linear regression models, as above. Confounding participant characteristics (i.e., age, sex, self-identified race, and years of education) were included and retained in all final models. Finally, to investigate the effects of divided attention on TUG performance (i.e., dual-task "costs"), we employed matched-pair t-tests across conditions for each of the nine mobility scores. Programming was done in SAS v.9.4 for Linux (SAS Institute Inc., Cary, NC) and R software [20]. P-values surviving Bonferroni correction ($p < .0055$) were considered significant.

## Results

Table 1 summarizes the participant, cognitive and overall TUG performance characteristics for the 757 adults included in these analyses. Under normal and undistracted testing, a longer TUG duration was associated with higher odds of MCI (OR per SD = 1.23 [1.04, 1.46], $p = .01$), lower global cognitive abilities ($\beta = -0.19$, $p < .001$), and poorer performance across the three cognitive domains ($-0.25 < \beta < -0.09$, $p's < .01$). However, when TUG was performed with the serial subtraction task, overall duration was longer (Cohen's $d = .88$, $p < .001$) and more strongly associated with MCI (OR = 1.38 [1.22, 1.56], $p < .001$), global cognition ($\beta = -0.31$, $p < .001$), and individual cognitive domains ($-0.26 < \beta < -0.22$, $p's < .001$).

### Associations of TUG subtasks with MCI

When we examined the relation of the nine mobility scores to MCI during the usual TUG, only slower walking pace and smaller turn magnitude were associated with higher odds of being diagnosed with MCI (Fig 1). We then considered overall task duration and all mobility scores together in a forward selection model to determine the best TUG indicators of MCI. Walking pace alone was selected by this procedure and accounted for a 2% increase in AUC beyond participant characteristics (OR = 0.71 [0.58,0.86], $p < .001$).

For the DT TUG, 6 of 9 mobility scores were associated with higher odds of MCI (Fig 1). Walking pace was again identified as the sole predictor of MCI in forward selection procedures

**Table 1. Participant, cognitive, and overall TUG performance characteristics.**

|  | All | NCI | MCI | NCI vs. MCI |
|---|---|---|---|---|
|  | (n = 757) | (n = 520) | (n = 237) | p-value |
| Age (years) | 81±7 | 80±7 | 84±7 | < .001 |
| Female (%) | 77.1 | 78.3 | 74.7 | .30 |
| Black (%) | 32.9 | 33.6 | 31.5 | .61 |
| Education (years) | 15.7±3.3 | 15.7±3.3 | 15.9±6.3 | .57 |
| **Cognitive Performance** |  |  |  |  |
| Mini-Mental State Exam | 28.1±1.8 | 28.5±1.4 | 27.1±2.2 | < .001 |
| Global Composite (z-score) | 0.24±0.27 | 0.43±0.46 | -0.17±0.55 | < .001 |
| **TUG Performance** |  |  |  |  |
| TUG Duration (s) | 15±7 | 14±6 | 17±8 | < .001 |
| Gait speed (m/s) | 0.77±0.3 | 0.81±0.3 | 0.67±0.3 | < .001 |
| DT TUG Duration (s) | 21±11 | 19±8 | 26±13 | < .001 |
| Gait speed (m/s) | 0.52±0.2 | 0.57±0.2 | 0.44±0.2 | < .001 |
| DT Serial Subtraction |  |  |  |  |
| Total # Responses | 7.2 ± 2.4 | 7.33 ± 2.2 | 6.9 ± 2.6 | .038 |
| # Correct | 6.2 ± 2.6 | 6.4 ± 2.4 | 5.6 ± 2.7 | < .001 |
| # Incorrect | 1.0 ± 1.4 | 0.9 ± 1.3 | 1.3 ± 1.4 | < .001 |

but was more strongly associated with MCI (OR = 0.53 [0.42, 0.66], $p < .001$) and accounted for a 4% increase in AUC beyond participant characteristics.

## Associations of TUG subtasks with global cognition

We then examined TUG subtask performance with a continuous composite metric of global cognition. For the normal TUG, 6 of 9 mobility scores were associated with the global cognitive composite (Fig 2). Walking pace and step time variability were selected as the best TUG

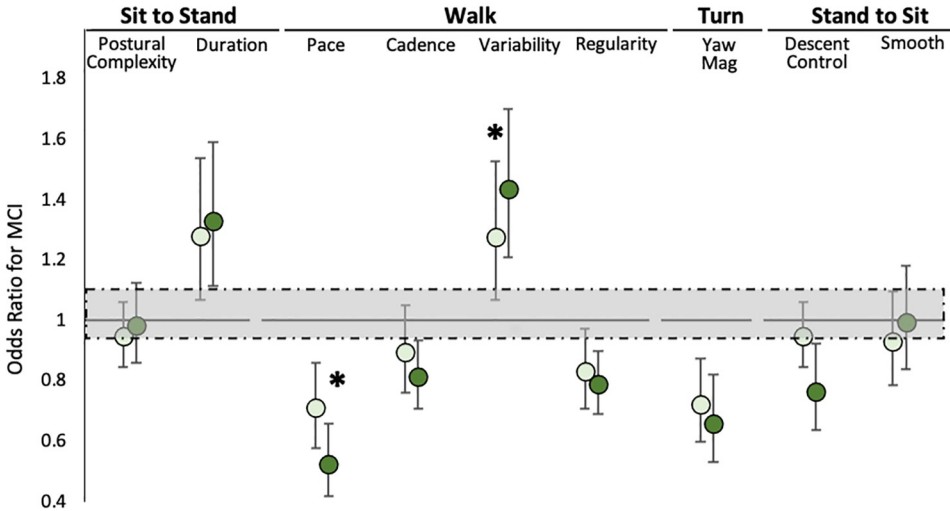

**Fig 1. Quantitative TUG vs. MCI.** Odds ratios and 95% confidence intervals for MCI based on individual mobility performance scores during normal (light green) and dual-task (dark green) Timed Up and Go (TUG). All models were adjusted for age, sex, race (Black versus White), and education. Statistically significant associations ($p < .005$) do not touch the shaded box. * indicates associations that significantly differ across conditions.

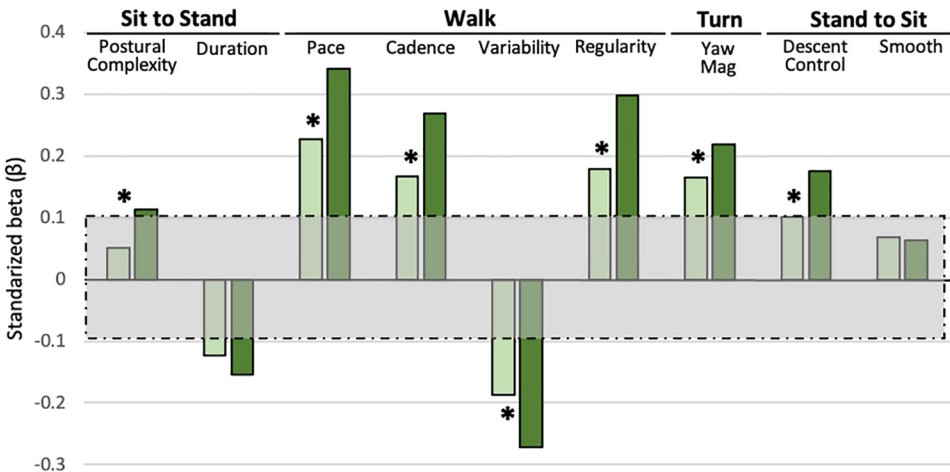

**Fig 2. Quantitative TUG vs. global cognition.** TUG subtasks vs. the global cognition composite, adjusted for age, sex, race, and education. Nine mobility scores quantifying the four TUG subtasks were calculated for normal (light green) and dual-task (dark green) TUG. A standardized beta (β) beyond the [-0.10, 0.10] shaded box corresponds to $p < .005$, while a β beyond [-0.133, 0.133] corresponds to $p < .0001$. * indicates associations that significantly differ across conditions.

indicators of global cognitive abilities and accounted for 5.4% and <0.5% of the variance, respectively.

For the DT TUG, 8 of 9 mobility scores were associated with global cognition. Walking pace alone was retained after forward selection but accounted for 11.7% of the variance (as compared to 5.4% without divided attention). Fig 2 illustrates subtask associations with the global composite and their differences across conditions.

## Associations of TUG subtasks with specific cognitive abilities

Next, we examined the associations of normal TUG with composites scores for executive functioning, attention, and episodic memory (Fig 3a). All 9 TUG subtask mobility scores were associated with executive functioning. Walking, turn, and stand-to-sit descent control metrics were associated with attention. Only walking pace and regularity were associated with episodic memory. In forward selection models, walking pace was the only TUG metric consistently associated with all three cognitive abilities. Pace and other metrics explained 6.7% of the variance in attention, 6% of executive functioning, and 3.6% of the episodic memory composite.

As expected, TUG associations with individual cognitive abilities were stronger with divided attention (Fig 3b) and associations with episodic memory emerged for 7 out of 9 mobility scores. Pace together with other dual-task TUG metrics explained 8% of the variance in attention, and 10.5% variance of executive function. Only 1.5% of the variation in episodic memory was explained by individual subtasks; 10% was accounted for by overall task duration. Associations with the traditional domains of our cognitive testing battery [21] may be found in S3 Table.

## Effects of divided attention on TUG performance

Finally, we examined the individual dual-task costs to isolate the effect of divided attention, i.e., TUG interference by the distractor task, and account for individual differences in motor

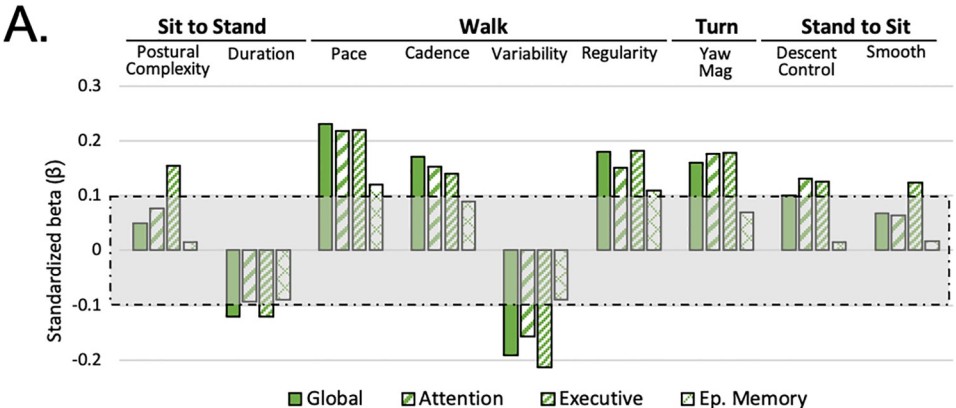

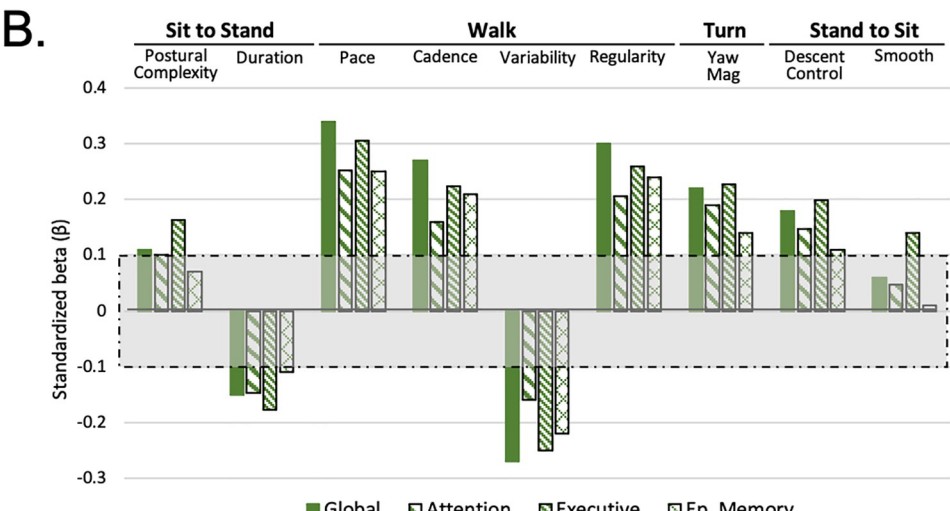

**Fig 3. Quantitative TUG vs. individual cognitive domains.** Associations between TUG subtasks and individual cognitive abilities during (a) normal and (b) dual-task (dark green) conditions, after adjusting for age, sex, race, and education. A standardized beta (β) beyond the [-0.10, 0.10] shaded box corresponds to $p < .005$, while a β beyond [-0.133, 0.133] corresponds to $p < .0001$.

abilities. Dividing attention during the TUG was associated with a small decrease in postural complexity (Cohen's $d$ = -0.161, $p < .001$) and increase in duration ($d$ = 0.25, $p < .001$) during the sit-to-stand transition, large decreases in pace ($d$ = -1.13, $p < .001$), cadence ($d$ = -1.09, $p < .001$), regularity ($d$ = -0.85, $p < .001$), and increased step time variability ($d$ = 0.64, $p < .001$) during walking. The magnitude of the turn in the middle of the trial was also substantially decreased ($d$ = -0.82, $p < .001$) along with a small decrease in descent postural control ($d$ = -0.22, $p < .001$) during the final stand-to-sit transition. That is, while participants tended to transition from sitting to standing more cautiously, they walked more slowly, walked and turned more irregularly, and sat more abruptly at the end of the dual-task protocol. Fig 4 illustrates the size of these effects across each of the nine mobility scores, such that the greatest effects of divided attention were observed during the walking and turning procedures and mirrored associations with global cognition. Individual quantitative metrics are also contrasted in S4 Table.

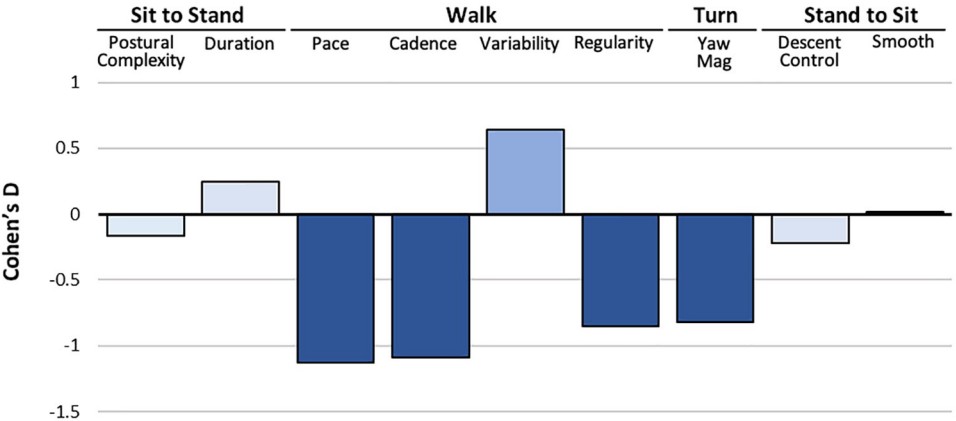

**Fig 4. Dual-task cost effect sizes during the four TUG subtasks.** Small effects of divided attention testing (Cohen's d<0.5; light blue) were observed in the control and duration of the sit-to-stand transition. Medium (>0.5) to large effects (>0.8; dark blue) were observed in the walking and turn procedures. A small effect was observed during the stand-to-sit transition only for postural descent control.

## Discussion

We employed an unobtrusive wearable sensor to quantify TUG performance with and without divided attention in a large diverse group of dementia-free older adults within the community. We first confirmed prior work that some TUG subtasks were associated with MCI and global cognition, then observed that all TUG subtasks became more strongly linked to cognition when performed under divided attention. We further observed that the degree to which an individual subtask was associated with cognitive abilities, especially attention and executive function, mirrored the degree to which it was altered by divided attention [22]. These context-dependent performances and associations throughout the TUG warrant further investigation of how other dual-tasking paradigms might be combined with quantitative mobility assessments to unmask motor and cognitive impairments in normal-appearing older adults.

To date, the majority of studies on the cognitive correlates and consequences of divided attention on motor performance has focused on gait [23, 24]. Slower, less accurate gait performance has been linked to underlying cognition at the time of assessment [25, 26], particularly attention and executive functioning, two domains crucial for walking and any goal-directed behaviors [5]. These associations with cognitive function are then made stronger with divided attention [27]. Since greater cognitive resources are required to ensure that two different simultaneous tasks are performed correctly, gait performance may beworsened, especially when these resources are limited. As a result, physical safety is typically prioritized over other tasks, such that even self-aware adults tend to walk more slowly or stop altogether [28–30] in order to keep balance and avoid harm. These same actions often foretell adverse outcomes in older age [31, 32]. Nonetheless, poorer multi-tasking abilities remain evidence of greater cognitive dysfunction.

TUG motor performances outside of gait remain largely underexplored. While task duration may be no more informative than gait speed in predicting geriatric outcomes [33], deconstructing the TUG could improve the information to be gained. Prior studies have linked compromised postural control and fluency during sit-to-stand transitions and turns with MCI and poor executive functioning [6, 34, 35]. Outside of these studies, however, we know of no publications examining instrumented TUG and cognition, especially during divided attention.

The current study suggests that the TUG may be highly informative of cognitive functioning, although clinical insight likely depends upon both subtask and context (e.g., with or

without divided attention). First, we report differential associations with TUG performance and cognition across the standing, walking, turning, and sitting procedures, such that walking and turning were most strongly coupled with cognitive outcomes and walking pace (i.e., gait speed) alone best informed cognitive status and abilities. We then provide evidence that dividing attention during the TUG can leverage its inherent associations with attention and executive functioning to increase difficulty across *several* task performances. That is, while attentional control is necessary to focus, select, and inhibit stimuli to prioritize neural resources toward a given task, and executive functioning is necessary to maintain task set, facilitate task shifts, monitor motor-sensory input, and safely interact with environment [22], these abilities become particularly important when both are needed to perform two concomitant tasks. And, although walking was indeed the most informative subtask towards cognition, dividing attention increased both the task difficulty and dependence on higher-level resources throughout the TUG procedure. These findings suggest that dual tasking may be used to uncover even subtle links between cognition and mobility.

When investigating interference by the unrelated serial subtraction task [36], we observed dual-task costs that paralleled subtask associations with cognition and differing levels of automaticity across the TUG. To the extent that divided attention costs were present but not strongly associated with cognition (i.e., postural transitions), we postulate that some movements are confounded by acts of caution, motor impairments, or are inherently more variable within individuals (e.g., gait variability across a short distance) [28–30]. That is, while costs in gait or turns offer insight towards cognitive outcomes, transitions may be better suited to predict adverse health outcomes like future frailty, falls, or disability in older adults [18].

The primary strengths of this study include our implementation of the instrumented dual-task TUG in a large, well-characterized racially diverse group of older adults in the community setting. We further contribute to existing literature by summarizing complex temporospatial gait and trunk kinematics into interpretable mobility scores via PCA, assessing cognitive-motor association effect sizes across different features of the task, and comparing these associations across conditions. To the best of our knowledge, we are also the first to observe dual-task costs that mirror associations with attentional and executive function abilities at the sub-movement level.

There are three main limitations. First, since we did not collect serial subtraction performance without TUG beyond a short practice, we were unable to assess cognitive "cost" or the relative task difficulty across normal and dual-task conditions. Second, without simultaneous recordings of verbal responses across the TUG, we were unable to ensure continuous serial subtraction task engagement or gauge task prioritization across participants. That is, since our divided attention task may not have affected all four motor subtasks to the same extent, the serial subtraction task perhaps best enhanced the cognitive correlates of ongoing motor execution, rather than earlier or later phases of the TUG. Additional testing paradigms are needed to better identify and delineate the cognitive and motor resources employed during other aspects of motor control, like movement planning, initiation, and termination. Finally, since this analysis does not consider longitudinal observations, it is not possible to infer the causal direction of the motor-cognitive associations which were observed. Further work will be needed to determine to what extent the instrumented DT TUG may improve the prediction of future cognitive impairment or other non-cognitive adverse health outcomes.

## Conclusions

In conclusion, our study demonstrates that instrumented dual-task testing increases the information to be gained from the TUG. While we found walking and turning to be most strongly

associated with cognitive abilities and impairment, all subtasks became more strongly linked to cognition when TUG was performed under divided attention. Motor performances were also altered by dual-tasking to the extent a subtask was associated with attention and executive function. These findings suggest dividing attention can be used to uncover both the shared and unshared resources across motor and cognitive tasks. Future work should consider similar paradigms to deconstruct complex movements, further probe the correlates of motor planning and execution, and better simulate everyday mobility and its associations with cognitive status and risk.

## Supporting information

**S1 Table. Attention and executive function composites.** Neuropsychological assessments used to compute composite scores of cognitive abilities across attentional and executive control domains. As in our prior publications, composite scores were constructed by averaging across centered and scaled test scores, relative to the appropriate parent study baseline. (DOCX)

**S2 Table. Mobility score components.** Measures extracted from each TUG performance were summarized as nine mobility scores to quantify four TUG subtasks. Some measures were flipped according to factor direction (e.g., AP jerk), denoted by a -1 multiplier. (DOCX)

**S3 Table. Quantitative TUG regression models for five cognitive domains.** Univariate and forward selection linear regression results for the five cognitive domains traditionally used to construct global cognition. Models feature TUG performances, adjusted for age, sex, race (Black versus White), and education. Please note that $\beta > 0.10$ has significance $p < 0.005$, while $\beta > 0.133$ has significance $p < .0001$. (DOCX)

**S4 Table. Quantitative TUG metrics across normal and dual-task conditions.** Individual metrics used to construct the nine mobility scores, as collected during the normal and dual-task Timed Up and Go. (DOCX)

## Acknowledgments

We thank all the MAP and MARS participants, and the staff and investigators at the Rush Alzheimer's Disease Center (RADC) for providing and processing high quality data. Please visit the RADC Research Resource Sharing Hub (www.radc.rush.edu) to obtain data for research purposes.

## Author Contributions

**Conceptualization:** Victoria N. Poole, Jeffrey M. Hausdorff, Aron S. Buchman.

**Data curation:** Robert J. Dawe, Sue E. Leurgans.

**Formal analysis:** Victoria N. Poole, Robert J. Dawe, Melissa Lamar.

**Funding acquisition:** Lisa Barnes, David A. Bennett, Aron S. Buchman.

**Methodology:** Robert J. Dawe, Sue E. Leurgans, Aron S. Buchman.

**Project administration:** Lisa Barnes, David A. Bennett, Aron S. Buchman.

**Software:** Robert J. Dawe, Jeffrey M. Hausdorff.

**Supervision:** David A. Bennett, Aron S. Buchman.

**Validation:** Robert J. Dawe, Melissa Lamar, Sue E. Leurgans.

**Visualization:** Victoria N. Poole.

**Writing – original draft:** Victoria N. Poole.

**Writing – review & editing:** Robert J. Dawe, Melissa Lamar, Michael Esterman, Lisa Barnes, Sue E. Leurgans, David A. Bennett, Jeffrey M. Hausdorff, Aron S. Buchman.

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
