## [Decision Letter · Decision Letter 0]

9 Mar 2022

PONE-D-21-36468Dividing attention during the Timed Up and Go enhances associations of subtask performances with MCI and cognitionPLOS ONE

Dear Dr. Poole,

Thank you for submitting your manuscript to PLOS ONE. After careful consideration, we feel that it has merit but does not fully meet PLOS ONE’s publication criteria as it currently stands. Therefore, we invite you to submit a revised version of the manuscript that addresses the points raised during the review process.

As you can see from the reviewers responses, there is substantial enthusiasm for this manuscript.  There are however several areas that will benefit from additional clarification.  It is important that the authors address the concerns of the reviewer with respect to state of the art versus novelty, keeping in mind that novelty is not a requirement for publication in PLOS ONE.  Adding this information to the manuscript will better place this manuscript within the existing body of literature.

We look forward to receiving your revised manuscript.

Kind regards,

Eric R. Anson

Academic Editor

PLOS ONE

Journal Requirements:

Reviewers' comments:

Reviewer's Responses to Questions

**Comments to the Author**

1. Is the manuscript technically sound, and do the data support the conclusions?

Reviewer #1: Yes

Reviewer #2: Partly

2. Has the statistical analysis been performed appropriately and rigorously? 

Reviewer #1: I Don't Know

Reviewer #2: Yes

3. Have the authors made all data underlying the findings in their manuscript fully available?

Reviewer #1: Yes

Reviewer #2: No

4. Is the manuscript presented in an intelligible fashion and written in standard English?

Reviewer #1: Yes

Reviewer #2: Yes

5. Review Comments to the Author

Reviewer #1: Dear Authors:

This is a very interesting study very pleasant to read and within a very pertinent and current area of interest.

The main strengths of this study is the implementation of the instrumented TUG with and without divided attention and thorough cognitive assessment in a large, well-characterized racially diverse group of older adults.

In my opinion the main limitations were also conveniently addressed.

Leave only the suggestion to consider the inclusion of a last subsection referring to the main conclusions of this study.

Reviewer #2: The paper is interesting and well written. The methodology is clear, and the analysis conducted thoroughly. However, I found some tricky points that, in my opinion, need clarification. Below I list the correction, doubts, and suggestions I have. I will divide them into general and specific arguments.

General comments:

1) as aforementioned said, the paper is clear and rigorous. Still, I have some concerns about the novelty. I would ask the authors to highlight the novelty of the work (if any) in terms of protocol, experimental design, kinematics/statistical analysis, or results. It is not clear what is scientific advancement compared to the state of the art.

2) State of art and related works are not sufficiently addressed.

3) It seems that the majority part of the paper results refers to the TUG walking subtask. Therefore, I wonder why the authors adopted the TUG as the motor activity rather than the commonly used walk task?

Specific Comments:

1) line 130, paragraph "Clinical Diagnosis of MCI and dementia": it is not clear if the test used served for the clinical diagnosis, research purposes, or both.

2) line 151, paragraph “Mobility Testing and Recording”: referring the sentence "…contained a tri-axial accelerometer and three gyroscopes..." is it possible that the authors meant a three-axis gyroscope instead of three gyroscopes?

3) lines 155-161, paragraph "TUG subtask sensor metrics": I found this part a bit convoluted. I would suggest the authors to make it clearer.

4) line 163, paragraph "TUG subtask sensor metrics": referring to the sentence "…by calculating the motor "cost..."" it seems that the scientific literature refers to this concept as “Dual-Task Cost (DTC)”. If it is so, I would suggest the authors to comply with this.

5) line 183, paragraph "Result": I would recommend the author to report the result consistently: e.g. line 190, there is no p-value associated to beta intervals. Check the section and homogenize the way you present the results, please.

6) line 249 section "Association of TUG subtasks with global cognition": I am wondering if beta levels should not be reported as an absolute value.

6. PLOS authors have the option to publish the peer review history of their article (what does this mean?). If published, this will include your full peer review and any attached files.

Reviewer #1: No

Reviewer #2: No

---

## [Author Response · Author response to Decision Letter 0]

27 Apr 2022

We would like to thank the editor and reviewers for the thoughtful critique of our manuscript and opportunity to submit a revision to PLOS ONE. We believe the manuscript is markedly improved after adding a concluding paragraph that highlights our study’s takeaways (Reviewer 1), and better clarifying its approach, novelty, and context within the existing literature (Reviewer 2). We have also reformatted the manuscript, in accordance with the PLOS requirements. Our specific responses to each review are emphasized below.

Response to Reviewers

and 

>> Thank you for sharing. We believe the manuscript now meets PLOS ONE’s style requirements.

Reviewer #1: Dear Authors:

This is a very interesting study very pleasant to read and within a very pertinent and current area of interest.

The main strengths of this study include the implementation of the instrumented TUG with and without divided attention and thorough cognitive assessment in a large, well-characterized racially diverse group of older adults.

In my opinion the main limitations were also conveniently addressed.

Leave only the suggestion to consider the inclusion of a last subsection referring to the main conclusions of this study.

>> Thank you for the kind feedback. We now include a final paragraph that summarizes our main conclusions.

Lines 389-398 now read:

“In conclusion, our study demonstrates that instrumented dual-task testing increases the information to be gained from the TUG. While we found walking and turning to be most strongly associated with cognitive abilities and impairment, all subtasks became more strongly linked to cognition when TUG was performed under divided attention. Motor performances were also altered by dual-tasking to the extent a subtask was associated with attention and executive function. These findings suggest dividing attention can be used to uncover both the shared and unshared resources across motor and cognitive tasks. Future work should consider similar paradigms to deconstruct complex movements, further probe the correlates of motor planning and execution, and better simulate everyday mobility and its associations with cognitive status and risk.”

Reviewer #2: The paper is interesting and well written. The methodology is clear, and the analysis conducted thoroughly. However, I found some tricky points that, in my opinion, need clarification. Below I list the correction, doubts, and suggestions I have. I will divide them into general and specific arguments.

>> Thank you for the careful review and suggestions.

General comments:

1) as aforementioned said, the paper is clear and rigorous. Still, I have some concerns about the novelty. I would ask the authors to highlight the novelty of the work (if any) in terms of protocol, experimental design, kinematics/statistical analysis, or results. It is not clear what is scientific advancement compared to the state of the art.

>> Thank you. We believe this work is novel for several reasons.

First, whereas very few studies have been able to deconstruct the TUG into its individual subtasks, we have deployed a wearable device able to quantify its spatiotemporal gait and postural movements in a large number of well-characterized older adults outside of clinic and lab, in the community setting. We further summarized these complex movements into interpretable mobility scores via PCA and report novel cognitive-motor associations across subtasks and conditions. Finally, to the best of our knowledge, we are the first to observe dual-task costs that mirrored associations with attentional and executive function abilities at the sub-movement level.

These contributions are now listed as study strengths on Lines 364-371.

2) State of art and related works are not sufficiently addressed.

>> Thank you. We now emphasize our combined use of the state-of-the-art McRoberts’ Dynaport MoveTest device with subsequent principal component analyses as critical to summarize the complex TUG into interpretable motor performances.

Lines 160-161: “This state-of-the-art technology allowed us to quantify both spatiotemporal gait and trunk kinematics during the TUG procedure in the community setting.”

Lines 169-171: “We then used principal component analyses (PCA) to reduce these [18 quantitative measures] into interpretable subtask performances, similar to our prior work.”

3) It seems that the majority part of the paper results refers to the TUG walking subtask. Therefore, I wonder why the authors adopted the TUG as the motor activity rather than the commonly used walk task?

>> Thank you for this question. By focusing on the TUG task, we were able to build upon our prior work and test the hypothesis that dual-tasking would enhance our ability to detect MCI from various movements throughout the TUG procedure. We were also able to determine the extent to which the individual subtasks depended on higher-level cognitive resources. Although walking was indeed the most informative subtask towards cognition, we believe it also noteworthy that even relatively automated sub-movements (i.e., transitions) were more closely coupled with cognition when performed under divided attention. These findings suggest that dual tasking may also be used to uncover otherwise subtle links between cognition and mobility (Lines 340-353).

Specific Comments:

1) line 130, paragraph "Clinical Diagnosis of MCI and dementia": it is not clear if the test used served for the clinical diagnosis, research purposes, or both.

>> Line 144 now states: “These diagnostic determinations were used for research purposes only.”

2) line 151, paragraph “Mobility Testing and Recording”: referring the sentence "…contained a tri-axial accelerometer and three gyroscopes..." is it possible that the authors meant a three-axis gyroscope instead of three gyroscopes?

>> Thank you for this question. The reviewer is correct.

Lines 157-159 now read: “This device contained a triaxial micro-electro-mechanical systems (MEMS) accelerometer and gyroscopic sensor to record acceleration and rotation, respectively, of the lower trunk at 100 Hz along each of three orthogonal directions.”

3) lines 155-161, paragraph "TUG subtask sensor metrics": I found this part a bit convoluted. I would suggest the authors to make it clearer.

>> Thank you for this suggestion.

Lines 166-173 now read: “Each continuous TUG trial recording was used to calculate the overall duration as well as 18 quantitative metrics, according to custom-developed algorithms. These metrics were transformed when appropriate to reduce skewness, averaged across trials for each condition, and z-scored relative to pooled parent study performance on the normal TUG. We then used principal component analyses (PCA) to reduce these data into interpretable subtask performances, similar to our prior work. Since the underlying factor structures were similar across normal and dual-task TUG, we created the same nine mobility scores for each condition, according to S2 Table.”

We have also tried to improve the clarity and flow throughout the manuscript.

4) line 163, paragraph "TUG subtask sensor metrics": referring to the sentence "…by calculating the motor "cost..."" it seems that the scientific literature refers to this concept as “Dual-Task Cost (DTC)”. If it is so, I would suggest the authors to comply with this.

>> Thank you. We now explicitly refer to motor costs as “dual-task costs” to better complement the literature.

5) line 183, paragraph "Result": I would recommend the author to report the result consistently: e.g. line 190, there is no p-value associated to beta intervals. Check the section and homogenize the way you present the results, please.

>> Thank you. All p-values are now reported.

6) line 249 section "Association of TUG subtasks with global cognition": I am wondering if beta levels should not be reported as an absolute value.

>> Thank you, we now include ranges (e.g., [-.10, .10]) to better highlight the different directions of association.

---

## [Decision Letter · Decision Letter 1]

20 May 2022

Dividing attention during the Timed Up and Go enhances associations of several subtask performances with MCI and cognition

PONE-D-21-36468R1

Dear Dr. Poole,

We’re pleased to inform you that your manuscript has been judged scientifically suitable for publication and will be formally accepted for publication once it meets all outstanding technical requirements.

Kind regards,

Eric R. Anson

Academic Editor

PLOS ONE

Additional Editor Comments (optional):

Reviewers' comments:

Reviewer's Responses to Questions

**Comments to the Author**

1. If the authors have adequately addressed your comments raised in a previous round of review and you feel that this manuscript is now acceptable for publication, you may indicate that here to bypass the “Comments to the Author” section, enter your conflict of interest statement in the “Confidential to Editor” section, and submit your "Accept" recommendation.

Reviewer #1: All comments have been addressed

2. Is the manuscript technically sound, and do the data support the conclusions?

Reviewer #1: Yes

3. Has the statistical analysis been performed appropriately and rigorously? 

Reviewer #1: Yes

4. Have the authors made all data underlying the findings in their manuscript fully available?

Reviewer #1: Yes

5. Is the manuscript presented in an intelligible fashion and written in standard English?

Reviewer #1: Yes

6. Review Comments to the Author

Reviewer #1: (No Response)

7. PLOS authors have the option to publish the peer review history of their article (what does this mean?). If published, this will include your full peer review and any attached files.

Reviewer #1: **Yes: **Catarina Godinho

---

## [Editor Report · Acceptance letter]

26 Jul 2022

PONE-D-21-36468R1 

Dividing attention during the Timed Up and Go enhances associations of several subtask performances with MCI and cognition 

Dear Dr. Poole:

I'm pleased to inform you that your manuscript has been deemed suitable for publication in PLOS ONE. Congratulations! Your manuscript is now with our production department. 

Kind regards, 

on behalf of

Dr. Eric R. Anson 

Academic Editor

PLOS ONE